# FeDeRA: Efficient Fine-tuning of Language Models in Federated Learning Leveraging Weight Decomposition

## Abstract

Federated learning (FL) is a widely used privacy-preserving approach for distributed training that avoids the need to collect data from individual users. In this paper, we investigate fine-tuning pre-trained language models (PLMs) in an FL setting and leverage parameter-efficient fine-tuning (PEFT) methods to reduce computational and communication costs. However, non-IID data in federated learning significantly degrades the performance of PEFT, with the degradation worsening as data heterogeneity increases. To address this, we propose FeDeRA, an FL approach for fine-tuning PLMs that incorporates an effective extension of the low-rank adaptation (LoRA) method. Specifically, FeDeRA initializes the low-rank matrices using Singular Value Decomposition (SVD) on the pre-trained weight matrices, rather than the zero or random initialization used in the original LoRA method. Analyzing weight updates during training reveals that FeDeRA reduces weight oscillations, enabling faster and more efficient fine-tuning of PLMs in FL with non-IID data. Experimental results across multiple NLP tasks and models show that FeDeRA outperforms all PEFT-based baselines in task performance and, in some cases, even matches or exceeds the performance of full-parameter fine-tuning. FeDeRA also greatly enhances training efficiency, reducing training time by up to 97.3% compared to full-parameter fine-tuning and up to 74.6% compared to the fastest PEFT baseline in practical FL settings. Furthermore, FeDeRA demonstrates greater robustness to data heterogeneity than all other PEFT methods, highlighting the effectiveness of its proposed initialization in FL systems.

## 1 Introduction

Pre-trained language models (PLMs) have achieved state-of-the-art (SOTA) performances across various NLP tasks, including natural language understanding(Kenton & Toutanova, 2019; Sanh et al., 2019; Liu et al., 2019), text generation(Brown et al., 2020; Zeng et al., 2022; Jiang et al., 2023), and question answering. However, training these models from scratch demands a significant amount of resources(Narayanan et al., 2021), making it unattainable for most. Consequently, fine-tuning models for specific tasks has emerged as the primary method of leveraging large language models. This process typically involves training a pre-trained model on a significantly smaller dataset rather than the original training set. While fine-tuning in a centralized manner is typically preferred, aggregating all data onto a single device raises concerns about data privacy, thus making centralized training increasingly challenging.

Federated learning(FL) (Konečnỳ et al., 2016; McMahan et al., 2017; Wang et al., 2022; Tang et al., 2022) has emerged as a promising approach in machine learning to address data privacy concerns by training a model collaboratively across decentralized clients without sharing raw local data. In FL, clients periodically compute and send model information, such as parameters or gradients, to a central server for aggregation, resulting in a global model. However, fine-tuning PLMs in an FL setting encounters significant challenges. Firstly, FL requires frequent exchange of model parameters or gradients between the server and clients. The massive number of parameters in PLMs, often in the tens or hundreds of billions, leads to substantial communication overheads. Additionally, devices involved in FL often communicate over bandwidth-limited networks, causing significant delays during data transmission and reducing training efficiency. Moreover, fine-tuning language

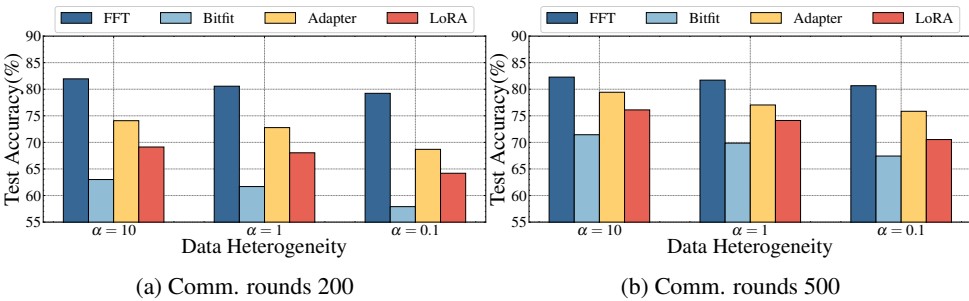

Figure 1: The performance of adopting PEFT methods within an FL setting is evaluated at varying levels of data heterogeneity using DistilBERT fine-tuned on the 20Newsgroup dataset. Heterogeneous data is generated based on a Dirichlet distribution, where the hyperparameter $\alpha$ determines the degree of data heterogeneity—a lower $\alpha$ value indicates higher data heterogeneity(Lin et al., 2022).

models demands substantial memory and computational resources, which many edge devices may not adequately meet.

Recently, parameter-efficient fine-tuning (PEFT) techniques such as BitFit(Zaken et al., 2022), adapter tuning(Narayanan et al., 2021; Pfeiffer et al., 2021), prefix tuning(Li & Liang, 2021) and LoRA(Hu et al., 2021) have garnered significant attention for their ability to update only a small portion of a pre-trained model's parameters, thus offering advantages in memory and computational efficiency. Existing works have demonstrated that these methods match or even surpass the performance of traditional full-parameter fine-tuning (FFT) methods in centralized training. Adopting PEFT in FL setting holds promise in addressing the aforementioned challenges by effectively reducing the number of parameters transmitted between clients and the server, as well as lowering the memory and computation costs associated with model training on each client.

In this work, we investigate PEFT methods within an FL setting and find that the heterogeneity of clients' data results in reduced performance and slower convergence rates for PEFT. As shown in Figure 1, the performance gap between PEFT methods and FFT widens as data heterogeneity increases. To address this issue, we propose FeDeRA, building upon LoRA's approach of decomposing pre-trained model weight matrices into low-rank matrices. FeDeRA innovatively leverages singular value decomposition (SVD) on pre-trained weight matrices to extract principal components for initializing low-rank matrices, i.e., the adaptors, while preserving and freezing remaining components in the original matrices. This simple yet effective solution addresses the remarkable performance decline observed in LoRA within federated settings, particularly when confronted with significantly non-IID data across clients. Experimental outcomes reveal that FeDeRA achieves comparable or superior task performance to FFT across various tasks, significantly surpassing other PEFT methods. Moreover, FeDeRA reduces training time by over 95% compared to FFT, while maintaining consistent task performance. Furthermore, experimental results underscore the robustness of FeDeRA against data heterogeneity, exhibiting much lower performance degradation compared to other PEFT methods.

## 2 RELATED WORKS

### 2.1 FEDERATED LEARNING WITH NON-IID DATA

FL is widely adopted for distributed learning, especially in tasks requiring significant privacy considerations. However, it suffers unneglectable performance degradation when data across different clients are non-independently and Identically Distributed (non-IID). Research efforts have surged to mitigate the effects of non-IID data on FL performance(Li et al., 2020). Some studies address this issue directly on the training data of each client through methods such as sharing a small portion of local data(Zhao et al., 2018), sharing model outputs(Itahara et al., 2021), enhancing data with external datasets(Jeong et al., 2018), generating new data from encoded and protected data of other clients(Shin et al., 2020), or selecting data carefully(Wang et al., 2020). Other approaches involves carefully designing the model training process, such as using adaptive learning rates on clients(Ma et al., 2021) or employing a control variable to prevent excessive drift in local updates(Karimireddy et al., 2020). Additionally, some studies have combined FL with other algorithms like Meta-Learning(Jiang et al.,

2019; Zhang et al., 2021), Lifelong Learning(Shoham et al., 2019; Kopparapu & Lin, 2020), and Knowledge Distillation(Itahara et al., 2021; Zhang et al., 2022). Existing research suggests that using pre-trained models in FL, rather than training models from scratch, can effectively mitigate the impact of non-IID data on performance(Weller et al., 2022; Nguyen et al., 2023; Chen et al., 2023). However, as discussed in section 1, the substantial computational demands and communication overheads associated with PLMs are significant challenges. Thus, optimizing the fine-tuning of PLMs within an FL setting for both computation and communication is urgently needed.

## 2.2 PEFT in Federated Learning

PEFT methods aim to freeze most or all of the weights in PLMs, fine-tuning only a small subset or newly introduced weights. This approach reduces computational resource requirements and significantly decreases communication overhead. Some studies have incorporated various PEFT methods in FL, demonstrating their efficiency in training and evaluating the comparative advantages and disadvantages of different PEFT approaches in terms of performance, resource demands, and privacy aspects(Sun et al., 2022; Chen et al., 2022; Zhang et al., 2023). Additionally, some other studies focus on exploring the potential of PEFT techniques within FL, such as using LoRA for enhanced heterogeneous personalized learning in FL(Yi et al., 2023; Lu et al., 2024), and incrementally enhancing adapter configurations during FL training to accelerate convergence(Cai et al., 2023). However, it has been increasingly recognized that PEFT methods experience significant performance degradation when dealing with highly non-IID data distributions in FL(Zhang et al., 2023; Babakniya et al., 2023). Despite this growing awareness, most prior research has either overlooked this critical issue or failed to evaluate PEFT methods under severe non-IID data conditions.

## 3 Preliminaries

### 3.1 Federated Learning

FL is a distributed approach to collaboratively train a global model across multiple clients, denoted by $\mathcal{M} = [1, 2 \ldots M]$, coordinated by a central server. We use $\mathcal{D}_m$ to denote the local training set at client $m \in \mathcal{M}$, and $|\mathcal{D}_m|$ to represent the number of training samples in $\mathcal{D}_m$. The goal of FL is to optimize the global model parameters $\omega$ by minimizing the following objective function:

$$\min_{\omega} F(\omega) \triangleq \sum_{m=1}^{M} p_m F_m(\omega), \tag{1}$$

where $p_m \geq 0$ is the weight assigned to the client $m$, with $\sum_{m=1}^{M} p_m = 1$. Typically, $p_m = \frac{|\mathcal{D}m|}{\sum m=1^M |\mathcal{D}_m|}$, ensuring that each training sample contributes equally to the optimization process. Additionally, $F_m(\cdot)$ represents the local objective, which can be expressed as:

$$F_m(\omega) \triangleq \mathcal{L}_m(\omega; \mathcal{D}_m), \tag{2}$$

where $\mathcal{L}_m(\cdot)$ is the specific local loss function at client $m$.

One of the most commonly used methods to solve this problem is the well-known FedAvg algorithm (McMahan et al., 2017). Specifically, in each training round, the server selects a subset of clients, denoted as $\mathcal{S} \in \mathcal{M}$, to participate in the FL training. $S = |\mathcal{S}|$ represents the number of selected clients. For a selected client $s \in \mathcal{S}$, the local training process begins by initializing its local model using the global model parameters from the previous round, i.e., $\omega_s = \omega^{t-1}$. The client then updates its local model parameters $\omega_s$ using its local dataset according to the following rule:

$$\omega_s \leftarrow \omega_s - \eta \nabla \mathcal{L}_s(\omega_s; \mathcal{D}_s), \tag{3}$$

where $\eta$ is the learning rate. This process is repeated for a specified number of iterations, resulting in the updated local weights $\omega_s^t$.

The selected clients then upload their updated weights, $\omega_s^t$, to the server. The server aggregates these received model parameters to update the global model parameters as follows:

$$\omega^t \leftarrow \frac{\sum_{s \in \mathcal{S}} |\mathcal{D}_s| \omega_s^t}{\sum_{s \in \mathcal{S}} |\mathcal{D}_s|}. \tag{4}$$

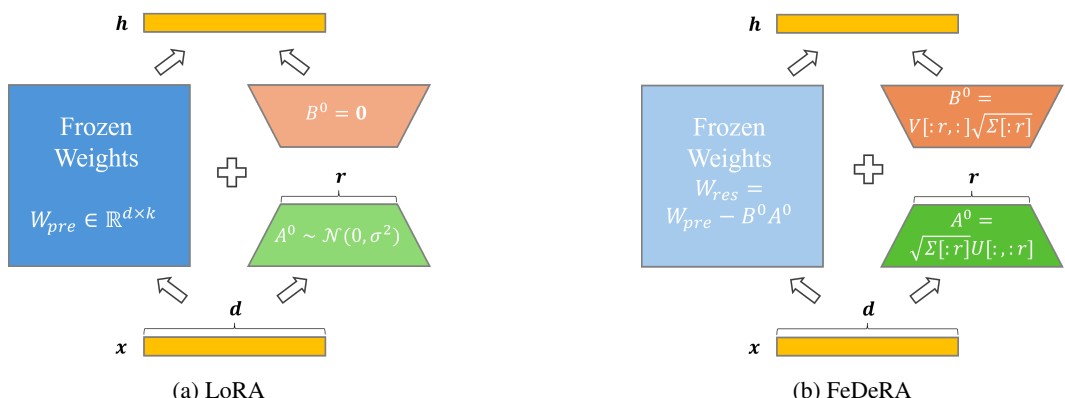

Figure 2: Weight initialization of LoRA and FeDeRA, $W_{pre} \in \mathbb{R}^{d \times k}$ denotes the pre-trained weights.

However, it is important to note that in real-world scenarios, the data across clients are often highly non-IID. This can lead to significant divergence among the local model parameters from different clients, making it challenging for the global optimization to converge.

## 3.2 LOW-RANK ADAPTATION

LoRA (Hu et al., 2021) is one of the most widely used PEFT methods of PLMs. This method is based on the observation that when PLMs are adapted to specific tasks, their weight update matrices usually have a lower intrinsic dimension than their dimensions. This implies that projecting these matrices into a lower dimension does not lead to significant information loss. The key idea behind LoRA is to avoid fine-tuning an entire pre-trained weight matrix $W_{pre} \in \mathbb{R}^{d \times k}$. Instead, as illustrated in Figure 2a, LoRA introduces a weight update mechanism based on matrix decomposition: $W_{pre} + \Delta W = W_{pre} + AB$. Here, $B \in \mathbb{R}^{d \times r}$ and $A \in \mathbb{R}^{r \times k}$ are trainable matrices, with rank $r \ll \min\{d, k\}$. During training, matrix $A$ is initialized by sampling from a random Gaussian distribution, and matrix $B$ is initialized to zeros, ensuring that initially $AB = 0$, as illustrated in Fig. 2a. The matrix $W_{pre}$ remains frozen throughout the training, while $AB$ are updated during the fine-tuning process. The output of a layer after implementing LoRA is given by:

$$h = W_{pre}x + \frac{\beta}{r}BAx, \tag{5}$$

where $x$ is the input, and $\beta$ is designed to eliminate the need of re-adjusting hyperparameters when changing $r$.

## 4 METHODS

### 4.1 FEDERA

We begin by analyzing why directly applying LoRA in FL suffers from a slower convergence rate and degraded performance when the data across clients is highly non-IID. Assuming a calculation order for a layer with a LoRA module is $Y = X(W_0 + AB) = XW_0 + XAB$. and assuming the update step is $t$, the update method for the $A$ and $B$ matrix is as follows:

$$A_t = A_{t-1} - \eta X_{t-1}^T \frac{\partial L_{t-1}}{\partial Y_{t-1}} B_{t-1}^T, \tag{6}$$

$$B_t = B_{t-1} - \eta A_{t-1}^T X_{t-1}^T \frac{\partial L_{t-1}}{\partial Y_{t-1}}. \tag{7}$$

According to the recurrence relation, let $Z_t = X_t^T \frac{\partial L_t}{\partial Y_t}$, we can obtain:

$$A_1 = A_0 - \eta Z_0 B_0^T, \tag{8}$$

$$B_1 = B_0 - \eta A_0^T Z_0. \tag{9}$$

Then we can derive

$$A_2 = A_1 - \eta Z_1 B_1^T$$
$$= A_0 - \eta(Z_0 + Z_1)B_0^T + \eta^2 Z_1 (A_0^T Z_0)^T, \tag{10}$$

$$B_2 = B_1 - \eta A_1^T Z_1$$
$$= B_0 - \eta A_0^T (Z_0 + Z_1) + \eta^2 (Z_0 B_0^T)^T Z_1. \tag{11}$$

Considering the initialization method in the original LoRA approach, i.e., $B_0 = 0$, we can recursively obtain:

$$A_t = A_0 + \eta^2 \Sigma_{i=1}^{t-1} (Z_i \Sigma_{j=0}^{i-1} Z_j^T) A_0^T, \tag{12}$$

$$B_t = -\eta A_0^T \Sigma_{i=0}^{t-1} Z_i. \tag{13}$$

Note that $Z_t$ is jointly contributed by the selected clients $\mathcal{S}_t$ in round $t$. In highly non-IID scenarios, due to the randomness in client selection and the client drift in gradients, the gradients obtained in each round may be biased in a different direction compared to the previous round. Hence, due to the zero and random initialization, as well as the double summation of gradients from previous rounds, i.e., $\Sigma_{i=1}^{t-1}(Z_i \Sigma_{j=0}^{i-1} Z_j^T)$, the weights of $A$ are updated more slowly, while the weights of $B$ experience significant oscillations. This makes it difficult for the model to begin converging at the start of training, slows down overall convergence, and can even degrade the model's performance.

To address this issue, we introduce FeDeRA, a novel FL approach for fine-tuning PLMs based on LoRA, which carefully designs the initialization of $A$ and $B$. The procedure begins with the SVD decomposition of $W_p re$, which can be expressed as:

$$W_{pre} \overset{\text{SVD}}{=} U\Sigma V, \tag{14}$$

where

$$U = [u_1, u_2, \ldots, u_d] \in \mathbb{R}^{d \times d},$$
$$V = [v_1, v_2, \ldots, v_k]^T \in \mathbb{R}^{k \times k},$$

denote the matrices of left and right singular vectors, respectively, and

$$\Sigma = \text{diag}(\sigma_1, \sigma_2, \ldots, \sigma_k) \in \mathbb{R}^{d \times k}, \tag{15}$$

is the diagonal matrix containing the singular values, $\sigma_1, ..., \sigma_k$. We then initialize the weight matrices $A$ and $B$ as follows with rank $r$:

$$A^0 = \sqrt{\Sigma[: r]}V[:, : r] = [\sqrt{\sigma_1}v_1, \sqrt{\sigma_2}v_2, \ldots, \sqrt{\sigma_r}v_r]^T, \tag{16}$$

$$B^0 = U[: r, :]\sqrt{\Sigma[: r]} = [\sqrt{\sigma_1}u_1, \sqrt{\sigma_2}u_2, \ldots, \sqrt{\sigma_r}u_r]. \tag{17}$$

As illustrated in Figure To maintain the original model output unchanged, we adjust and freeze the initial matrix by:

$$W_{res} = W_{pre} - B^0 A^0. \tag{18}$$

### 4.2 ANALYSIS

We elucidate the superiority of FeDeRA by delving into the process of model updating. Firstly, define magnitude vector $M_w \in \mathbb{R}^{1 \times k}$ and direction matrix $D_w \in \mathbb{R}^{d \times k}$ for a weight matrix $W \in \mathbb{R}^{d \times k}$ as(Liu et al., 2024):

$$M_w = [m_w^1, m_w^2 \ldots m_w^k] = [||w^1||_2, ||w^2||_2 \ldots ||w^k||_2], \tag{19}$$

$$D_w = [d_w^1, d_w^2 \ldots d_w^k] = [\frac{w^1}{m_w^1}, \frac{w^2}{m_w^2} \ldots \frac{w^k}{m_w^k}], \tag{20}$$

where $w^j$ is the $j$-th column vector of $W$. Define magnitude variation $\Delta M_w(\cdot)$ and direction variation $\Delta D_w(\cdot)$ between two weights matrices $W_1 \in \mathbb{R}^{d \times k}$ and $W_2 \in \mathbb{R}^{d \times k}$ as:

$$\Delta M_w(W_1, W_2) = \frac{\Sigma_{n=1}^k |m_{w1}^n - m_{w2}^n|}{k}, \tag{21}$$

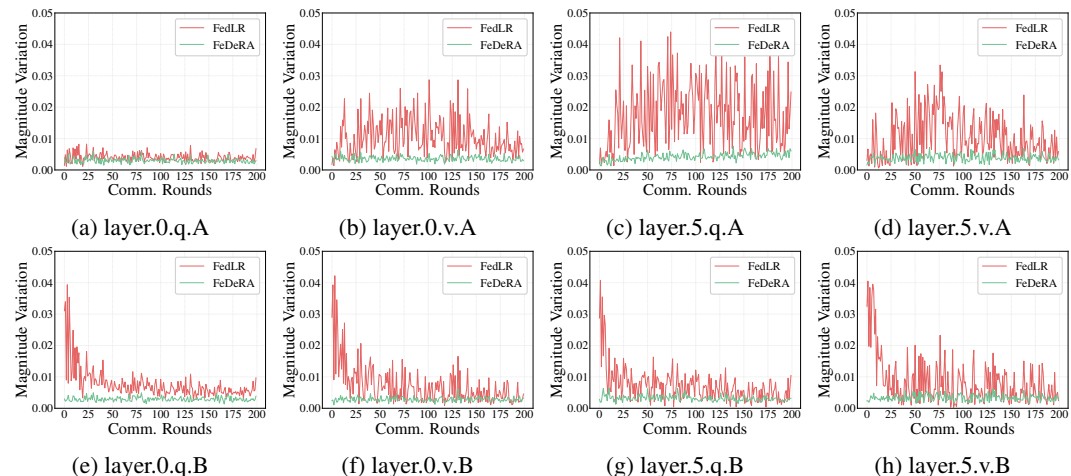

Figure 3: Magnitude variation in consecutive global weight updates by FeDeRA and FedLR fine-tuning DistilBERT over 200 Communication Rounds on the 20 Newsgroups Dataset.

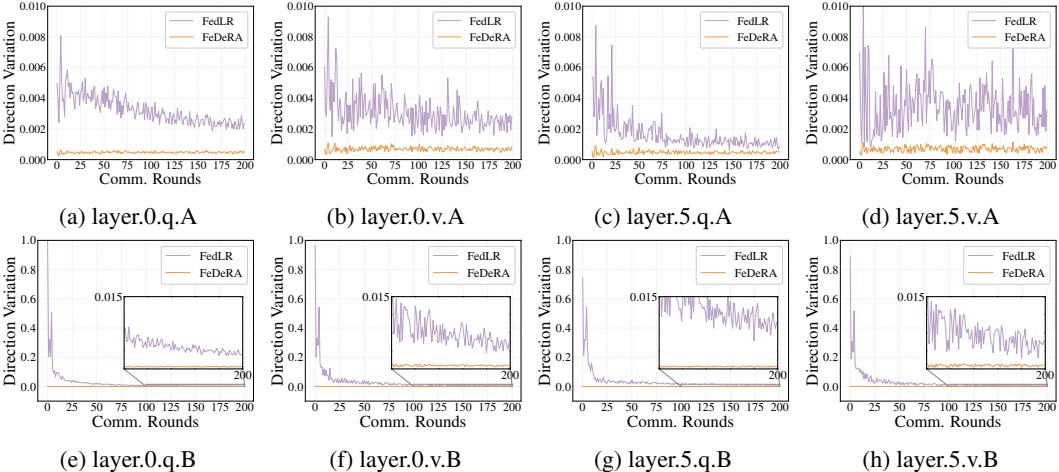

Figure 4: Direction variation in consecutive global weight updates by FeDeRA and FedLR fine-tuning DistilBERT over 200 communication rounds on the 20Newsgroups dataset.

$$\Delta D_w(W_1, W_2) = \frac{\Sigma_{n=1}^{k}(1 - \text{CosineSimilarity}(d_{w1}^n, d_{w2}^n))}{k}. \tag{22}$$

We then evaluate the magnitude and direction variations in successive global weight updates by FeDeRA and FedLR, referring to the original LoRA approach adapted to FL. Specifically, we calculate $\Delta M_w(\omega^t, \omega^{t-1})$ and $\Delta D_w(\omega^t, \omega^{t-1})$ for both FedLR and FeDeRA, as shown in Figure 3 for $t$ ranging from 0 to 200. The results indicate that FedLR exhibits more dramatic changes in the parameter updates of matrices $A$ and $B$, with magnitude and direction variations in successive updates being several to hundreds of times greater than those by FeDeRA. This is particularly noticeable for $B$ matrix at the beginning of the training, likely due to its initialization to zero. As illustrated in Figure 4e, Figure 4f, Figure 4g, and Figure 4h, the initial weight updates tend to be orthogonal or opposite to their consecutive updates. This supports our assertion that LoRA's initialization under highly heterogeneous data training leads to more severe weight update drift, making convergence more challenging. In contrast, our proposed FeDeRA methodology exhibits stability in both magnitude variation and direction changes, facilitating faster convergence and effectively mitigating the impact of data heterogeneity in FL.

## 5 EVALUATION

In this section, we present numerical results to evaluate the performance of the proposed FeDeRA using the FedML(He et al., 2020) and FedNLP(Lin et al., 2022) frameworks. Specifically, we first compare the performance of FeDeRA with various baselines across six datasets and three NLP tasks, using highly non-IID training data generated according to a Dirichlet distribution. We then implement FeDeRA and the baseline methods on practical devices and access time cost of different approaches to achieve a target accuracy to evalute the training efficiency.

### 5.1 BASELINES

We adopt existing FFT and PEFT methods within the FL setting as baselines. To ensure a fair comparison, the experimental settings for these baselines are consistent with those in their original works, unless explicitly stated otherwise. The details of the baselines are as follows:

**FedFT:** The conventional approach which updates all the model's parameters throughout the fine-tuning process.

**FedBF:** Each client updates only the bias terms while keeping the other parameters fixed.

**FedAP:** This approach incorporates adapter-tuning by inserting some trainable adapter layers into the model while keeping the parameters of the original model frozen. Specifically, adapters are inserted after the attention and feed-forward modules, following the methodology outlined in (Houlsby et al., 2019). The adapter module is implemented using the adapters library (Poth et al., 2023; Pfeiffer et al., 2020).

**FedLR:** This approach directly applies the original LoRA to FL. Note that the LoRA operation is applied only to the query and value matrices, while the key matrices are kept fixed. The same applies to the proposed FeDeRA. LoRA implementation is carried out using the PEFT library (Mangrulkar et al., 2022).

### 5.2 SETUP

**Models and Datasets.** We adopt RoBERTa-base (Liu et al., 2019) and DeBERTaV3-base (He et al., 2022) as the backbone models in our experiments, implementing them using the transformers library (Wolf et al., 2020). We consider three NLP tasks: text classification (TC), named entity recognition (NER), and question answering (QA). Specifically, we use the 20Newsgroups (Lang, 1995) and SemEval-2010Task8 (Hendrickx et al., 2010) datasets for text classification, WNUT2017(Derczynski et al., 2017) and PLONER(Fu et al., 2020) datasets for named entity recognition, SQuADv1.1(Rajpurkar et al., 2016) and MRQA(Fisch et al., 2019) datasets for question answering, respectively. Additional details about the datasets and parameter configurations can be found in Table 1. It is noteworthy that we select as small $\alpha$ as possible to ensure extreme data heterogeneity without significantly impeding the convergence of all methods. Further information on the choice of $\alpha$ is available in Appendix A.

Table 1: Details of datasets and Non-IID settings. $*$ indicates that the corresponding datasets are divided into 80% and 20% for training and testing, respectively. # of classes denotes the number of classes in each dataset. $\alpha$ is used to control the magnitude of data heterogeneity.

| Dataset | Task | #Train | #Test | # of clients | # selected per round | # of classes | $\alpha$ |
|---------|------|--------|-------|--------------|---------------------|--------------|----------|
| 20Newsgroup | Text Classification | 11.3k | 7.5k | 100 | 10 | 20 | 0.1 |
| SemEval-2010Task8 | Text Classification | 8k | 2.7k | 100 | 10 | 19 | 1 |
| WNUT | Named Entity Recognition | 3.4k | 1.2k | 30 | 5 | 37 | 0.1 |
| PLONER$^*$ | Named Entity Recognition | 14k | 3.5k | 100 | 10 | 49 | 0.01 |
| SQuADv1.1 | Question Answering | 87.5 | 34.7 | 300 | 15 | 30 | 0.1 |
| MRQA$^*$ | Question Answering | 45.5k | 11.3k | 300 | 15 | 6 | 0.01 |

**Training settings.** The training batch size is configured to 16, with each local training epoch set to 1. We select the best learning rate from $[1 \times 10^{-3}, 5 \times 10^{-4}, 1 \times 10^{-4}, 5 \times 10^{-5}, 1 \times 10^{-5}]$ for each backbone model and dataset setting. Additionally, the maximum sequence length is set to 64 for SemEval-2010Task8, WNUT2017, and PLONER, 256 for 20Newsgroup, 384 for SQuADv1.1, and

512 for MRQA. For FedLR and FeDeRA, the rank $r$ is set to 32, and $\beta$ equation 5 is set to be equal to $r$.

## 5.3 TASK PERFORMANCE

Table 2, Table 3 and Table 4 present the performance of FeDeRA and the baselines on TC, NER, and QA tasks. The results indicate that FeDeRA consistently outperforms FedBF, FedAP, and FedLR across these tasks. Moreover, FeDeRA achieves comparable performance to FedFT and even surpasses it in some cases, with only about 1% of trainable parameters. We also note that FedFT requires fewer training rounds to converge compared to all the PEFT-based methods including FedBF, FedAP, and FedLR. There is a substantial performance gap between these methods and FedFT at 200 training rounds, which only gradually narrows with additional training rounds. FeDeRA also demonstrates significantly faster convergence compared to these PEFT-based baselines. After more training rounds, it consistently matches the performance of FedFT, with only minor differences in the early stages. These results underscore the superiority of the proposed approach.

Table 2: Evaluation results on text classification tasks. The data marked with **bold** and underlined indicates the best and second-best results achieved by the five methods under each configuration. Each result is derived by averaging over five experiments with different random seeds. This applies to Table 3 and Table 4 as well.

| Model | Method | #Trainable Parameters | 20NEWS(Acc.) Comm. rounds | | | SEMEVAL(Acc.) Comm. rounds | | |
|---|---|---|---|---|---|---|---|---|
| | | | 200 | 500 | 1000 | 200 | 500 | 1000 |
| RoBERTa | FedFT | 125M | $\mathbf{79.56}_{\pm 0.6}$ | $\mathbf{82.71}_{\pm 0.2}$ | $\mathbf{83.67}_{\pm 0.3}$ | $\underline{80.34}_{\pm 1.6}$ | $\mathbf{82.99}_{\pm 0.4}$ | $\underline{83.53}_{\pm 0.8}$ |
| | FedBF | 0.1M | $65.41_{\pm 0.9}$ | $70.32_{\pm 0.3}$ | $72.21_{\pm 0.2}$ | $72.03_{\pm 1.1}$ | $77.28_{\pm 0.5}$ | $78.66_{\pm 0.1}$ |
| | FedAP | 1.8M | $74.73_{\pm 0.1}$ | $78.46_{\pm 0.3}$ | $80.78_{\pm 0.1}$ | $72.98_{\pm 0.5}$ | $79.39_{\pm 0.1}$ | $80.66_{\pm 0.2}$ |
| | FedLR | 1.2M | $73.17_{\pm 0.6}$ | $78.03_{\pm 0.2}$ | $80.25_{\pm 0.3}$ | $74.81_{\pm 1.7}$ | $79.26_{\pm 0.5}$ | $80.37_{\pm 0.8}$ |
| | **FeDeRA** | 1.2M | $\underline{77.24}_{\pm 0.2}$ | $\underline{80.33}_{\pm 0.2}$ | $\underline{82.21}_{\pm 0.1}$ | $\mathbf{80.74}_{\pm 0.6}$ | $\underline{82.91}_{\pm 0.5}$ | $\mathbf{83.78}_{\pm 0.4}$ |
| DeBERTaV3 | FedFT | 184M | $\underline{76.47}_{\pm 1.2}$ | $\underline{82.25}_{\pm 0.6}$ | $\underline{83.99}_{\pm 0.4}$ | $\mathbf{79.57}_{\pm 2.5}$ | $\mathbf{83.66}_{\pm 1.2}$ | $\mathbf{84.42}_{\pm 1.1}$ |
| | FedBF | 0.1M | $38.93_{\pm 1.3}$ | $55.72_{\pm 0.2}$ | $61.37_{\pm 1.3}$ | $31.44_{\pm 1.9}$ | $64.37_{\pm 0.9}$ | $73.79_{\pm 0.9}$ |
| | FedAP | 1.8M | $58.75_{\pm 1.7}$ | $73.54_{\pm 0.8}$ | $79.09_{\pm 0.4}$ | $64.05_{\pm 1.1}$ | $78.08_{\pm 0.7}$ | $80.46_{\pm 0.2}$ |
| | FedLR | 1.2M | $50.41_{\pm 1.3}$ | $73.02_{\pm 0.9}$ | $80.16_{\pm 0.2}$ | $39.76_{\pm 2.3}$ | $71.11_{\pm 1.9}$ | $80.13_{\pm 1.4}$ |
| | **FeDeRA** | 1.2M | $\mathbf{76.91}_{\pm 1.1}$ | $\mathbf{82.57}_{\pm 0.5}$ | $\mathbf{84.38}_{\pm 0.4}$ | $\underline{72.45}_{\pm 2.5}$ | $\underline{83.09}_{\pm 0.8}$ | $\underline{84.36}_{\pm 0.5}$ |

Table 3: Evaluation results on named entity recognition tasks .

| Model | Method | #Trainable Parameters | WNUT(F1) Comm. rounds | | | PLONER(F1) Comm. rounds | | |
|---|---|---|---|---|---|---|---|---|
| | | | 100 | 200 | 500 | 50 | 100 | 150 |
| RoBERTa | FedFT | 125M | $\mathbf{50.03}_{\pm .5}$ | $\underline{51.53}_{\pm .9}$ | $\underline{52.31}_{\pm .2}$ | $\mathbf{86.01}_{\pm .6}$ | $\mathbf{88.01}_{\pm .1}$ | $\mathbf{89.19}_{\pm .1}$ |
| | FedBF | 0.1M | $35.71_{\pm 2.9}$ | $42.45_{\pm .9}$ | $45.23_{\pm 1.1}$ | $75.95_{\pm .6}$ | $79.81_{\pm .1}$ | $80.78_{\pm .1}$ |
| | FedAP | 1.8M | $47.81_{\pm 1.1}$ | $50.11_{\pm .1}$ | $50.25_{\pm .2}$ | $79.23_{\pm .1}$ | $83.21_{\pm .6}$ | $85.72_{\pm .2}$ |
| | FedLR | 1.2M | $46.39_{\pm 2.4}$ | $49.71_{\pm .3}$ | $50.15_{\pm .6}$ | $81.14_{\pm .1}$ | $85.29_{\pm .1}$ | $86.88_{\pm .4}$ |
| | **FeDeRA** | 1.2M | $\underline{49.14}_{\pm 1.3}$ | $\mathbf{52.28}_{\pm .7}$ | $\mathbf{52.73}_{\pm .8}$ | $\underline{84.74}_{\pm .5}$ | $\underline{87.14}_{\pm .1}$ | $\underline{88.44}_{\pm .1}$ |
| DeBERTaV3 | FedFT | 184M | $\mathbf{50.26}_{\pm .7}$ | $\mathbf{51.04}_{\pm .8}$ | $\mathbf{51.45}_{\pm .6}$ | $\mathbf{81.41}_{\pm .4}$ | $\mathbf{86.24}_{\pm .3}$ | $\mathbf{87.06}_{\pm .1}$ |
| | FedBF | 0.1M | $42.79_{\pm .5}$ | $46.25_{\pm .3}$ | $47.54_{\pm .4}$ | $70.44_{\pm .1}$ | $75.22_{\pm .2}$ | $77.54_{\pm .3}$ |
| | FedAP | 1.8M | $43.29_{\pm 1.3}$ | $45.45_{\pm 1.9}$ | $47.68_{\pm .5}$ | $77.46_{\pm .2}$ | $82.69_{\pm .5}$ | $84.82_{\pm .6}$ |
| | FedLR | 1.2M | $43.75_{\pm 1.3}$ | $46.78_{\pm 1.7}$ | $48.82_{\pm .6}$ | $76.61_{\pm .3}$ | $82.55_{\pm .5}$ | $83.88_{\pm .2}$ |
| | **FeDeRA** | 1.2M | $\underline{50.02}_{\pm 1.2}$ | $\underline{50.51}_{\pm 1.1}$ | $\underline{51.14}_{\pm .6}$ | $\underline{81.02}_{\pm .2}$ | $\underline{84.59}_{\pm .1}$ | $\underline{86.11}_{\pm .1}$ |

## 5.4 TRAINING EFFICIENCY

We then evaluate the training efficiency of our proposed method in terms of time cost to reach a targeted accuracy in a practical scenario, where we use Jetson AGX Orin as clients, and a server equipped with 8×NVIDIA RTX A6000 GPUs and 2×64-Core AMD EPYC 7763 CPUs. These devices are connected via WiFi with bandwidth ranging from 20 to 30 Mbps. We measure the time cost of different methods to reach 90%, 95%, and 99% of the target metrics, as shown in **??**. From this table, we observe that FeDeRA significantly reduces the time required to achieve a specified accuracy

Table 4: Evaluation results on question answering tasks.

| Model | Method | #Trainable Parameters | SQuADv1.1(EM/F1) | | MRQA(EM/F1) | |
|---|---|---|---|---|---|---|
| | | | Comm. rounds | | Comm. rounds | |
| | | | 50 | 100 | 100 | 200 |
| RoBERTa | FedFT | 125M | $63.16_{\pm.3}$ / $77.84_{\pm.3}$ | $65.27_{\pm.2}$ / $79.62_{\pm.1}$ | $51.45_{\pm.5}$ / $60.78_{\pm.1}$ | $52.44_{\pm.2}$ / $63.24_{\pm.3}$ |
| | FedBF | 0.1M | $43.87_{\pm2.4}$ / $60.91_{\pm2.1}$ | $50.09_{\pm.7}$ / $66.53_{\pm.8}$ | $25.67_{\pm1.5}$ / $37.88_{\pm1.1}$ | $33.69_{\pm.1}$ / $45.55_{\pm.3}$ |
| | FedAP | 1.8M | $62.02_{\pm.1.2}$ / $76.84_{\pm1.2}$ | $65.14_{\pm.2}$ / $79.42_{\pm.4}$ | $47.38_{\pm.1}$ / $59.21_{\pm.3}$ | $51.01_{\pm.1}$ / $62.25_{\pm.4}$ |
| | FedLR | 1.2M | $57.37_{\pm.2}$ / $72.68_{\pm.1}$ | $60.87_{\pm.5}$ / $75.91_{\pm.4}$ | $46.85_{\pm.3}$ / $58.4_{\pm.2}$ | $49.88_{\pm.5}$ / $61.16_{\pm.3}$ |
| | **FeDeRA** | 1.2M | $62.53_{\pm.2}$ / $\underline{77.51}_{\pm.2}$ | $65.31_{\pm.3}$ / $79.98_{\pm.2}$ | $\underline{49.73}_{\pm.5}$ / $61.27_{\pm.4}$ | $\underline{52.05}_{\pm.3}$ / $\underline{63.11}_{\pm.5}$ |
| DeBERTaV3 | FedFT | 184M | $65.81_{\pm.3}$ / $81.12_{\pm.2}$ | $66.39_{\pm.3}$ / $81.15_{\pm.3}$ | $55.11_{\pm.2}$ / $66.46_{\pm.1}$ | $56.15_{\pm.1}$ / $67.34_{\pm.1}$ |
| | FedBF | 0.1M | $53.51_{\pm.4}$ / $71.32_{\pm.6}$ | $57.49_{\pm.1}$ / $74.81_{\pm.3}$ | $23.03_{\pm2.3}$ / $34.19_{\pm.6}$ | $30.72_{\pm1.4}$ / $43.34_{\pm.5}$ |
| | FedAP | 1.8M | $\underline{63.32}_{\pm.2}$ / $79.21_{\pm.1}$ | $65.35_{\pm.1}$ / $80.76_{\pm.1}$ | $52.32_{\pm.3}$ / $64.31_{\pm.1}$ | $55.56_{\pm.5}$ / $66.65_{\pm.3}$ |
| | FedLR | 1.2M | $61.05_{\pm.2}$ / $77.78_{\pm.3}$ | $63.84_{\pm.3}$ / $79.25_{\pm.2}$ | $50.45_{\pm.2}$ / $62.56_{\pm.2}$ | $53.71_{\pm.1}$ / $65.31_{\pm.1}$ |
| | **FeDeRA** | 1.2M | $63.24_{\pm.4}$ / $\underline{79.27}_{\pm.4}$ | $65.54_{\pm.4}$ / $80.59_{\pm.1}$ | $53.64_{\pm.1}$ / $65.96_{\pm.1}$ | $55.92_{\pm.4}$ / $67.14_{\pm.2}$ |

compared to all the baselines. This is because FeDeRA requires fewer training rounds to reach a certain accuracy than other PEFT-based methods. Although FeDeRA requires more training rounds to converge compared to FedFT, it offers substantial acceleration in both training and communication with far fewer trainable parameters. While FeDeRA introduces an additional SVD computation, this is performed only once and is negligible compared to the overall training time. Specifically, across different models and tasks with varying target accuracies, FeDeRA reduces training time by 76.9% to 97.3% compared to FedFT, and by 36.7% to 74.6% compared to the fastest PEFT-based method. These results demonstrate that FeDeRA can significantly improve training efficiency in practical FL systems.

Table 5: Comparison on training efficiency by different methods, measured by time required to achieve 90%, 95%, and 99% of the target metrics, which are accuracy of 0.8 for text classification on 20Newsgroup, F1 score of 0.5 for named entity recognition on WNUT2017, EM score of 0.65 and F1 score of 0.8 for question answering on SQuADv1.1. "-" indicates that the target metric cannot be achieved by the corresponding method under the considered settings. The time cost is measured in hours.

| Model | Method | 20NEWS(Acc.) | | | WNUT(F1) | | | SQuADv1.1(EM/F1) | | |
|---|---|---|---|---|---|---|---|---|---|---|
| | | 90% | 95% | 99% | 90% | 95% | 99% | 90% | 95% | 99% |
| RoBERTa | FedFT | 5.60 | 6.53 | 13.19 | 5.22 | 7.04 | 10.38 | 1.39 / 0.84 | 2.31/ 1.96 | 5.02/ 5.64 |
| | FedBF | 0.78 | - | - | 0.89 | - | - | - / - | - / - | - / - |
| | FedAP | 0.30 | 0.49 | 0.99 | 0.30 | 0.44 | 0.95 | 0.21 / 0.12 | 0.39 / 0.34 | 0.65 / 0.73 |
| | FedLR | 0.33 | 0.59 | 1.21 | 0.27 | 0.39 | 0.72 | 0.45 / 0.24 | 0.85 / 0.84 | 1.43 / 1.79 |
| | FeDeRA | **0.17** | **0.30** | **0.54** | **0.21** | **0.28** | **0.32** | **0.16 / 0.09** | **0.31 / 0.28** | **0.51 / 0.52** |
| DeBERTaV3 | FedFT | 13.27 | 16.91 | 23.84 | 6.48 | 9.14 | 12.96 | 0.67 / 0.48 | 0.96 / 0.67 | 1.62 / 1.34 |
| | FedBF | - | - | - | 0.63 | 1.27 | - | 1.13 / 0.50 | - / - | - / - |
| | FedAP | 1.13 | 1.74 | - | 0.69 | 1.40 | - | 0.22 / 0.15 | 0.36 / 0.25 | 0.67 / 0.49 |
| | FedLR | 1.03 | 1.43 | 1.80 | 0.81 | 0.85 | - | 0.33 / 0.18 | 0.55 / 0.34 | 1.12 / 0.77 |
| | FeDeRA | **0.36** | **0.45** | **0.64** | **0.16** | **0.32** | **0.45** | **0.14 / 0.09** | **0.24 / 0.19** | **0.49 / 0.31** |

## 5.5 IMPACT OF DATA HETEROGENEITY

In this section, we investigate the impact of non-IID on the training accuracy by changing the value of $\alpha$, as shown in Figure 5. We use RoBERTa-base model and the 20Newsgroup dataset. It can be observed that as $\alpha$ decreases, i.e., as data heterogeneity increases, the performance of all methods deteriorates. However, the performance of FedFT remains relatively stable compared to other approaches, demonstrating its robustness to data heterogeneity in an FL setting by updating all parameters. The proposed FeDeRA also experiences much less performance degradation compared to other methods, even with significantly fewer trainable parameters than FedFT. These results further demonstrate the effectiveness of FeDeRA in handling data heterogeneity.

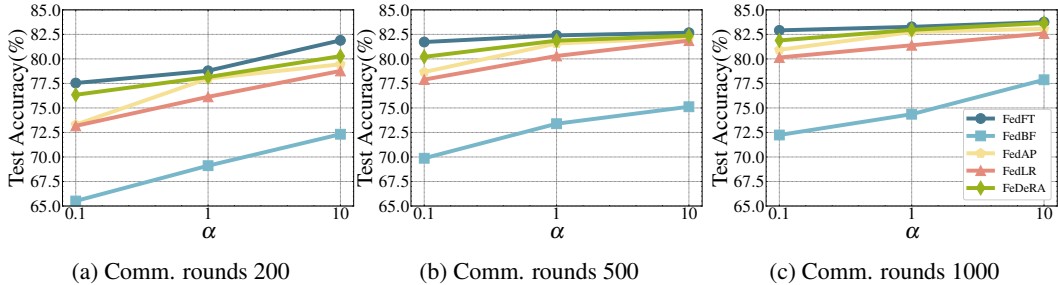

(a) Comm. rounds 200     (b) Comm. rounds 500     (c) Comm. rounds 1000

Figure 5: Performance comparison on test accuracy with regards to different levels of data heterogeneity.

## 5.6 IMPACT OF TRAINABLE PARAMETER BUDGETS

We further evaluate the performance of different methods in terms of model accuracy and training time with respect to various trainable parameter budgets, as shown in Figure 6.We use the RoBERTa-base model and the 20Newsgroup dataset, setting $\alpha$ to 1 and the total training rounds to 500 for the experiments presented in Figure 6a. As seen in Figure 6a, FeDeRA significantly outperforms the three benchmarks in terms of model accuracy across a wide range of trainable parameters, especially when the parameter budget is small. Additionally, in Figure 6b and Figure 6c, where we set the target accuracy to 0.76 and 0.8, respectively, we observe that the proposed FeDeRA also requires less training time under various trainable parameter budgets. This further demonstrates the effectiveness of FeDeRA.

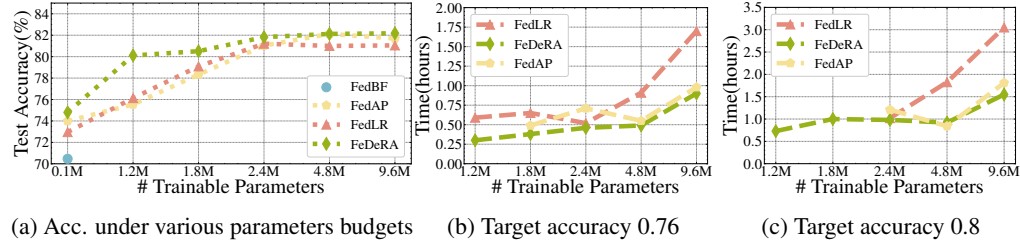

(a) Acc. under various parameters budgets    (b) Target accuracy 0.76    (c) Target accuracy 0.8

Figure 6: Performance comparison on test accuracy and training time cost with regards to different trainable parameter budgets.

## 6 CONCLUSIONS

In this paper, we introduced FeDeRA, a method for fine-tuning PLMs in a federated setting. Specifically, FeDeRA extends LoRA by initializing the low-rank adapters with the results of Singular Value Decomposition on the pre-trained weight matrices. We analyzed the weight update equations and visualized the magnitude and direction curves of weight updates during training, explaining how FeDeRA reduces weight oscillations, thereby accelerating and improving the fine-tuning of PLMs in an FL setting with non-IID data. Our empirical findings demonstrated that FeDeRA outperforms all PEFT-based baselines in terms of task performance and closely approximates, or even surpasses in some cases, the performance of the full-parameter fine-tuning approach. It was also shown that FeDeRA significantly improves training efficiency, reducing training time by up to 97.3% compared to the full-parameter fine-tuning baseline, and by up to 74.6% compared to the fastest PEFT-based baseline in a practical setting. Furthermore, FeDeRA is more robust to data heterogeneity than all other PEFT methods, demonstrating the effectiveness of the proposed initialization in FL systems.

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

# A NON-IID DATA PARTITION

## A.1 DIRICHLET DISTRIBUTION

The Dirichlet distribution, commonly referred to as the multivariate Beta distribution, is a class of high-dimensional continuous distributions. The support of this distribution is the standard simplex within the realm of positive real numbers. It is an extension of the Beta distribution to higher dimensions. The Dirichlet distribution, delineated by a parameter measure denoted as $\mathbf{u}$, can be represented through its corresponding probability density function as:

$$f(\boldsymbol{x}; \boldsymbol{u}) = \frac{1}{B(\boldsymbol{u})} \prod_{i=1}^{K} x_i^{u_i - 1},$$

where $B(\boldsymbol{u})$ is the multivariate beta function:

$$B(\boldsymbol{u}) = \frac{\prod_{i=1}^{K} \Gamma(u_i)}{\Gamma(\sum_{i=1}^{K} u_i)},$$

and $\Gamma(\cdot)$ denotes the Gamma function. $\boldsymbol{u}$ is typically expressed as $\boldsymbol{u} = \alpha\boldsymbol{m}, \sum_{i=1}^{K} m_i = 1, m_i > 0$. Note that for a random variable $X \sim Dir(\alpha\boldsymbol{m})$, $\boldsymbol{m}$ is the mathematic expectation of $X$:

$$\mathbb{E}(X) = \int f(\boldsymbol{x}; \boldsymbol{u})\boldsymbol{x} \, \mathrm{d}\boldsymbol{x} = \boldsymbol{m}.$$

The magnitude of $\alpha$ determines the degree of similarity between the sampling distribution and the original distribution; a larger $\alpha$ yields a sampling distribution that is more similar to the original.

## A.2 CLASSES OF DATASETS

We defined a specific number of classes for each dataset to facilitate the partition of the datasets. The following is the detailed information.

**20Newsgroup.** The 20Newsgroups dataset comprises around 18000 newsgroups posts on 20 topics and the topics cover a range of subjects from politics to religion to sports to science. The 20 Newsgroups dataset is employed for text classification tasks; consequently, its original labels are utilized as its classes.

**SemEval2010Task8.** SemEval2010-Task8 is aimed at addressing the challenge of Multi-Way Classification of Semantic Relations Between Pairs of Nominals. The task involves identifying and categorizing the semantic relations between specific pairs of nominals within given sentences. It delineates nine distinct types of semantic relations, with each relation type being subject to inversion, thereby totaling 19 potential labels when including an additional 'Other' category for relations that do not conform to any of the predefined types. We also utilize the original labels as classes.

**WNUT2017.** The WNUT2017 dataset refers to the data collection used for the Workshop on Noisy User-generated Text (WNUT) in its 2017 edition. This workshop series focuses on processing and understanding noisy user-generated text, often encountered in social media, online forums, and other web sources. WNUT2017 specifically aimed to address several key tasks in noisy text processing. One of the primary tasks was Named Entity Recognition (NER), which involved identifying and classifying proper names within text into predefined categories such as persons, organizations, locations, etc. The token labels comprise seven parts of speech, further divided into begin and intermediate positions, along with an additional category that does not belong to any of the aforementioned parts of speech, totaling thirteen labels in all. We observe that each training sample may comprise a unique set of token labels, thereby contributing to the learning capability across diverse token labels. Consequently, we catalogue the total types of token labels encompassed within each training sample as its class, as detailed in Table 6.

**PLONER.** PLONER, the abbreviation for Person, Location, Organization Named Entity Recognition, aims to assess the cross-domain generalization capability. It involves selecting samples from representative datasets that include at least one of the three entity types: person, location, and organization. The representative datasets encompass CoNLL2003, OntoNotes 5.O, and WNUT2016.

Table 6: Examples of constructing classes for the WNUT2017 dataset.

| Tokens | Token Labels | Class |
|---|---|---|
| [ "today", "is", "my", "last", "day", "at", "the", "office", "." ] | [ O, O, O, O, O, O, O, O, O ] | O |
| [ "Pxleyes", "Top", "50", "Photography", "Contest", "Pictures", "of", "August", "2010", "...", "http://bit.ly/bgCyZ0", "#photography" ] | [B-corporation, O, O, O, O, O, O, O, O, O, O ] | B-corporation |
| [ "Toy", "story", "3", "tonight", "on", "the", "lawn", "!" ] | [B-creative-work, I-creative-work ,I-creative-work, O, O, O, O, O ] | B-creative-work-I-creative-work |
| [ "(", "via", "POPSUGAR", ")", "Sarah", "Jessica", "Parker", "and", "Gwen", "Stefani", "Wrap", "Up", "Another", "Successful", "New", "York", "Fashion", "Week", ":", "New", "York", "Fa", "...", "http://bit.ly/aMaJNB" ] | [ O, O, O, O,B-person, I-person , I-person, O,B-person, I-person, O, O, O, O, O, O, O, O, O, B-location , I-location, O, O, O ] | B-person-I-person-B-location-I-location |

Table 7: Examples of constructing classes for the PLONER dataset.

| Tokens | Token Labels | Source Dataset | Class |
|---|---|---|---|
| [ "only", "France", "and", "Britain", "backend", "Fischler", "'s", "proposal", "." ] | [ O, B-LOC, O, B-LOC , O, B-PER, O, O, O ] | CoNLL2023 | CoNLL-B-LOC-B-PER |
| ["Cowboys", "on", "a", "3rd", "and", "10", "finally", "get", "their", "1st", "down", "in", "the", "game", "."] | [ B-ORG, O, O, O, O, O, O, O, O, O, O, O, O, O, O ] | WNUT2016 | WNUT-B-ORG |
| ["As", "you", "know", "Lebanon", "is", "an", "agreement", "-", "based", "democracy", "."] | [ O, O, O, B-LOC, O, O, O, O, O, O, O ] | OntoNotes5.0-BN | OnBN-B-LOC |
| ["In", "contrast", ",", "Taiwan", "alumni", "are", "noticeable", "by", "their", "absence", "from", "the", "SAR", "government", "in", "Hong", "Kong", "."] | [ O, O, O, B-LOC, O, O, O, O, O, O, O, O, B-ORG, I-ORG, I-ORG, O, B-LOC, I-LOC, O ] | OntoNotes5.0-MZ | OnMZ-B-LOC-I-LOC-B-ORG-I-ORG |

The English data of CoNLL2003 is a collection of news wire articles from the Reuters Corpus. The data for WNUT2016 were primarily collected from Twitter. The OntoNotes5.0 corpus is categorized into several types, which include newswire (News), broadcast news (BN), broadcast conversation (BC), telephone conversation (Tele), and web data (Web). The corpora across various themes inherently exhibit heterogeneity. Hence, while adopting the methodology used for class construction in WNUT2017, we also take into account the source dataset of the samples for further distinction, details are shown in Table 7.

**SQuADv1.1.** SQuADv1.1 is a question-answering dataset released by Stanford University. The dataset comprises an extensive collection of question-and-answer pairs, with the questions formulated based on paragraphs from Wikipedia articles, and the answers are extracted directly from these respective passages. SQuADv1.1 does not annotate any tags pertinent to the questions, hence, we employ k-means to generate classes for all training samples directly. The number of clusters chosen for this study is 30.

**MRQA.** The MRQA dataset emphasizes the generalizability of question-answering tasks. This dataset is an amalgamation of subsets from 18 existing QA datasets, carefully curated and transformed into the format as SQuAD. Of these 18 datasets, six are designated for training, another six for development, and the final six are reserved for testing purposes. In our approach, we simply employ the source datasets as the classes.

## A.3 Partitioning Non-IID Datas With Classes

We obtain the non-IID datasets for the clients by partitioning the original dataset based on the Dirichlet distribution and the classes we have defined. Let $C$ denote the total number of classes, and $c_i$ represent the total number of data instances for the $i$-th class in the undivided dataset. We compute the distribution for classes as $\boldsymbol{m_c} = [m_{c,1}, m_{c,2} \ldots m_{c,C}], m_{c,i} = \frac{c_i}{\sum_{i=1}^{C} c_i}$. Assuming there are N clients, we sample the local data class distribution $X_i = [x_{i,1}, x_{i,2} \ldots x_{i,N}]$ for the $i$-th client according to the Dirichlet distribution, denoted as $X_i \sim Dir(\alpha \boldsymbol{m_c})$, $x_{i,j}$ represents the proportion of the $j$-th class on the $i$-th client.

## A.4 Visualization of Data Heterogeneity

We visualized the data heterogeneity among different clients according to the data partitioning according to subsection A.3. We employed the Jensen-Shannon divergence to characterize the

disparities in the distributions of data across various clients:

$$JS(P||Q) = \frac{1}{2}\sum p_i \log \frac{p_i}{q_i} + \frac{1}{2}\sum q_i \log \frac{q_i}{p_i}$$

where $P$ and $Q$ denote two probability distributions defined in the same space. The value of Jensen-Shannon divergence ranges from 0 to 1, with larger values indicating a greater disparity between two distributions. Visualization results are presented in Figure 7, illustrating that the degree of non-IID with various values of $\alpha$.

## B  DETAILS OF TIME COST

Table 8 and Table 9 present the local training time and communication time for each training round in FL, as well as the time consumption for SVD decomposition in the experiments of Table 5 and Figure 6. It can be observed from Table 8 that FedFT has significantly larger training and communication overheads than other methods, with the communication cost being particularly higher compared to methods that incorporate PEFT. This is why, even though FedFT requires fewer training rounds and can achieve better performance, we are still interested in exploring PEFT in FL. Although FeDeRA introduces additional SVD time, it only needs to be performed once during the entire training process, which is almost negligible compared to the total training time. Table 9 illustrates that as the number of trainable parameters gradually increases, both the client's training and communication times rise incrementally, guiding us in selecting the optimal training configuration while considering the trade-off between training time cost and model performance.

Table 8: The local training time and communication time for each training round by different methods, and the time consumption for SVD decomposition, measured in seconds.

| Model | Dataset | FedFT | | FedBF | | FedAP | | FedLR&FeDeRA | | FeDeRA |
|---|---|---|---|---|---|---|---|---|---|---|
| | | Train | Comm. | Train | Comm. | Train | Comm. | Train | Comm. | SVD |
| **RoBERTa** | **20NEWS** | 4.59 | | 3.15 | | 3.29 | | 3.27 | | |
| | **WNUT** | 4.62 | 109.67 | 3.31 | 0.96 | 3.42 | 2.47 | 3.33 | 1.68 | 8.62 |
| | **SQuADv1.1** | 30.91 | | 21.83 | | 22.8 | | 22.37 | | |
| **DeBERTaV3** | **20NEWS** | 6.31 | | 4.36 | | 4.62 | | 4.54 | | |
| | **WNUT** | 6.22 | 149.98 | 4.49 | 0.96 | 4.81 | 2.47 | 4.62 | 1.68 | 8.62 |
| | **SQuADv1.1** | 42.44 | | 29.51 | | 31.86 | | 30.46 | | |

Table 9: The local training time and communication time for each training round with different number of trainable parameters by FeDeRA, measured in seconds.

| Method | Trainable Parameters | | | | | | | | | |
|---|---|---|---|---|---|---|---|---|---|---|
| | 0.1M | | 1.2M | | 1.8M | | 2.4M | | 4.8M | |
| | Tain | Comm. | Tain | Comm. | Tain | Comm. | Tain | Comm. | Tain | Comm. |
| FedBF | 3.15 | | - | - | - | - | - | - | - | - |
| FedAP | 3.31 | 0.96 | 3.25 | 1.68 | 3.29 | 2.47 | 4.05 | 3.24 | 4.17 | 4.55 |
| FedLR&FeDeRA | 3.14 | | 3.22 | | 3.26 | | 3.95 | | 4.04 | |

## C  ADDITIONAL MAGNITUDE AND DIRECTION VARIATION

We present the results of magnitude variation and direction variation not included in subsection 4.2 here, as shown in Figure 8 and Figure 9. As can be observed, our proposed FeDeRA demonstrates more stability in weight updates across all layers of DistilBERT and, consequently, converges more easily compared to FedLR.

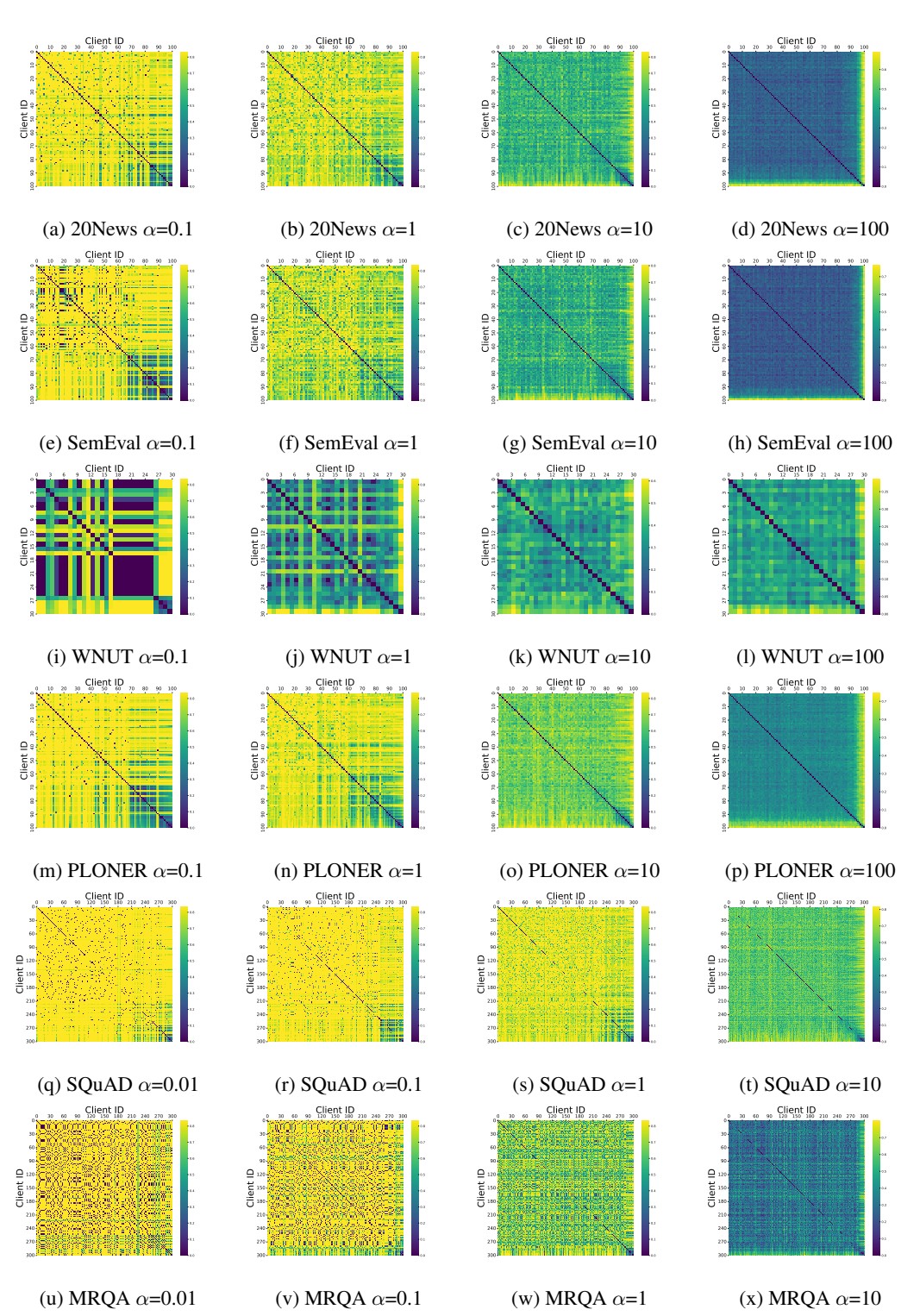

Figure 7: Visualization of data heterogeneity across six datasets for various values of $\alpha$, where lighter colors denote a larger Jensen-Shannon divergence, indicating greater disparity between the sampling distribution and the original data distribution

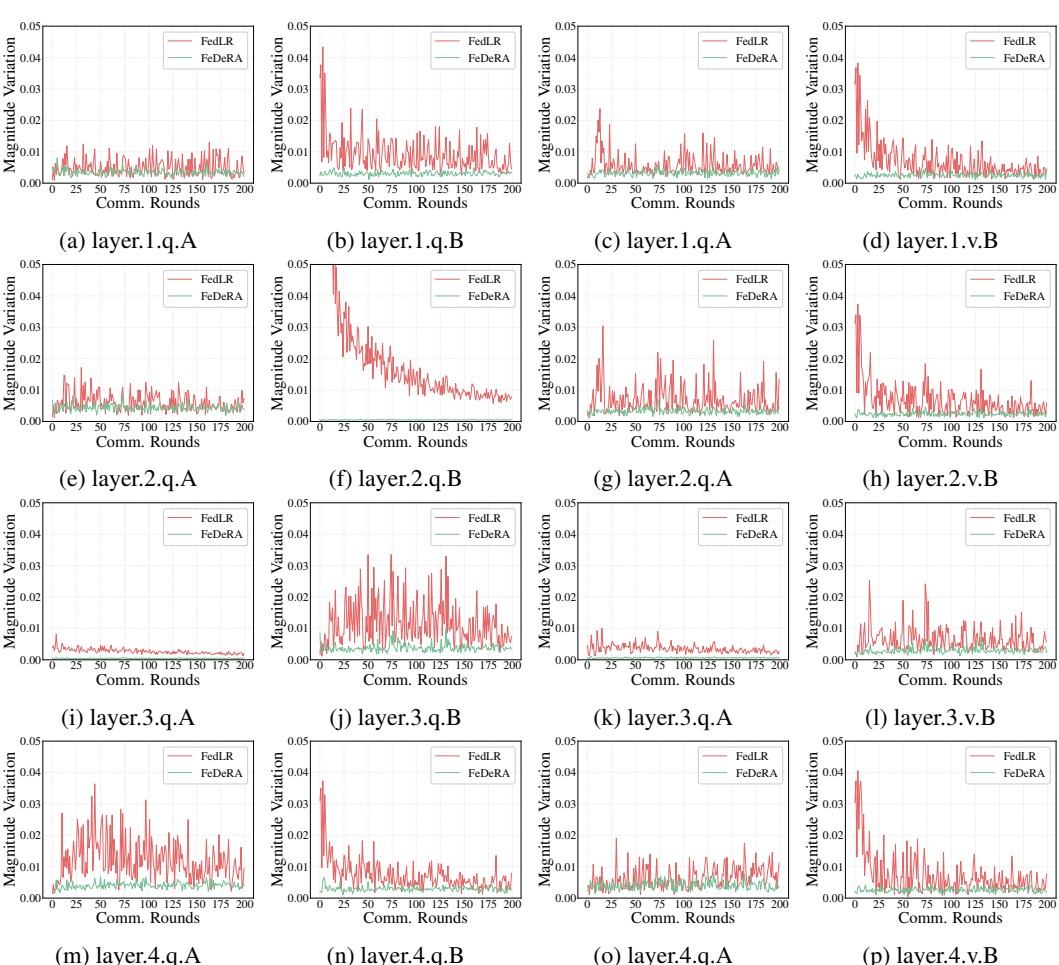

Figure 8: Additonal magnitude variation of DistilBERT over 200 Communication Rounds on the 20Newsgroups dataset.

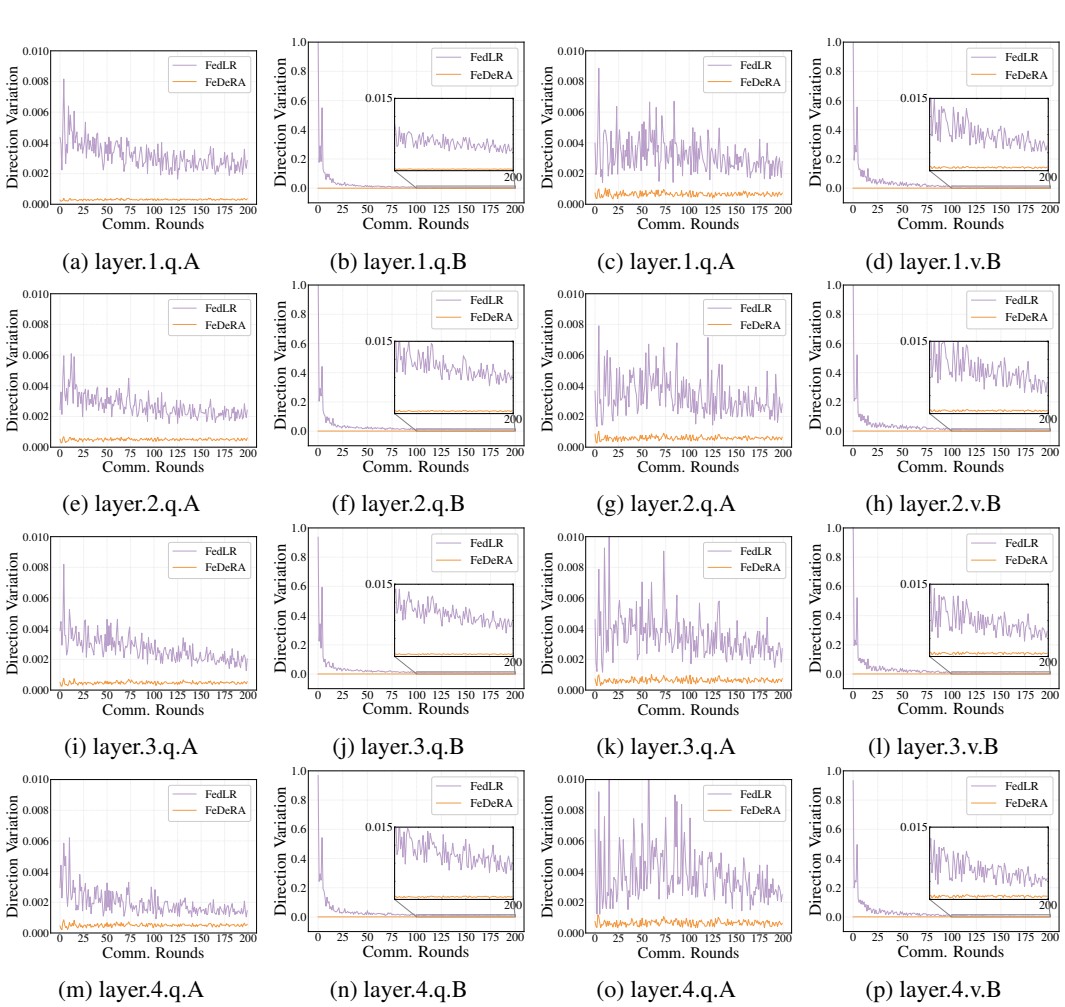

Figure 9: Additional direction variation of DistilBERT over 200 communication rounds on the 20Newsgroups dataset.

