# OpenReview forum: "FeDeRA: Efficient Fine-tuning of Language Models in Federated Learning Leveraging Weight Decomposition"
_ICLR.cc/2025/Conference — ICLR 2025 Conference Withdrawn Submission_

### Official Review · Reviewer_fDcY · 2024-11-03

**Soundness:** 2
**Presentation:** 3
**Contribution:** 2
**Rating:** 3
**Confidence:** 4

**Summary:**

This paper introduces FeDeRA (Federated Decomposition of Representations with Adaptation), a novel federated learning (FL) method designed to fine-tune pre-trained language models (PLMs). FeDeRA builds on the low-rank adaptation (LoRA) technique by performing singular value decomposition (SVD) on pre-trained weight matrices to initialize corresponding low-rank matrices, thereby enhancing the efficiency of parameter updates during fine-tuning. The method aims to address the challenges of communication efficiency and model performance in FL settings, particularly when data across clients is non-IID. Key contributions include the introduction of SVD-based initialization for low-rank matrices, a reduction in the number of trainable parameters, and empirical demonstrations of improved performance over existing PEFT methods and comparable performance to full parameter fine-tuning (FFT) with significantly reduced computational costs.

**Strengths:**

1.	Initializing the A and B matrices in LoRA with the SVD decomposition of pre-trained weight matrices is an interesting and novel approach.

2.	The analysis of magnitude variation and direction variation suggests that such an initialization can help reduce the variation in model updates, potentially leading to more stable and consistent training.

**Weaknesses:**

1.	The method mainly relies on performing SVD decomposition on pre-trained weight matrices to initialize the low-rank matrices, which may not be considered a substantial innovation

2.	Performing SVD on large pre-trained weight matrices can be computationally expensive, particularly when the weight matrices have very high dimensions, posing significant demands on computational resources.

3.	The purpose of adjusting W0←W0−BA is to ensure that the model output is unchanged at the beginning by outputting the model when A, B are not initialized to 0. However, since A and B are derived from the SVD of W0 in the paper, the new matrix values may be close to zero, which may cause the original information embedded in W0 to be lost.

4.	The paper claims that the method helps reduce the variation in model updates and improve performance under data heterogeneity. However, it does not provide a detailed explanation or theoretical justification for how and why this approach effectively addresses the challenges posed by data heterogeneity.

**Questions:**

Read the Weaknesses for more details.

---

### Official Review · Reviewer_GsaJ · 2024-11-04

**Soundness:** 2
**Presentation:** 2
**Contribution:** 1
**Rating:** 3
**Confidence:** 5

**Summary:**

The authors introduce the FeDeRA algorithm to reduce training and communication costs in federated learning. In this algorithm, the authors utilize the SVD of the pre-trained weights to initialize the LoRA blocks instead of randomly initializing them.

**Strengths:**

* The problem statement and motivation are clearly explained in the paper.

* Their method is evaluated on mutiple settings.

**Weaknesses:**

The main contribution of the paper is using SVD on the pre-trained weights, which is very incremental.

**Questions:**

Could the authors explain their contributions?

---

### Official Review · Reviewer_rzHJ · 2024-11-04

**Soundness:** 1
**Presentation:** 2
**Contribution:** 2
**Rating:** 3
**Confidence:** 4

**Summary:**

In this submission, the authors propose FeDeRA, a new method for federated fine-tuning based on LoRA.

**Strengths:**

The experiments are well-designed, different aspects (such as performance, efficiency and data heterogeneity) are covered. These results are promising. It would become more convincing by including more necessary baselines and model.

**Weaknesses:**

1) Experiment baselines are missing.
To validate the effectiveness of the proposed FeDeRA, it is necessary to experimentally compare with other federated fine-tuning methods such as [1] [2], and more in [3]. If these federated fine-tuning methods are not suitable for comparison, could the authors clearly state the reasons? Otherwise, the missing of very related baselines make the submission less convincing, as it fails to compare with existing related work.
[1] Slora: Federated parameter efficient fine-tuning of language models
[2] improving lora in privacy-preserving federated learning
[3] Federated Large Language Models: Current Progress and Future Directions

2) Discussion about related work is not sufficient.
As pointed above, there are plenty of existing work about federated fine-tuning of LLM. Thus in Section 2.2, it is necessary to discuss these related work and clearly state the technique novelty of the proposed FeDeRA compared to existing methods. In this way, readers can better understand the technique contribution of the proposed method.

3) More experiment settings can be necessary.
Authors only conduct experiments with RoBERTa-base (2019) and DeBERTaV3-base (2022). As the rapid development of LLM in recent two years, it is necessary to conduct experiments with modern LLM such as LLaMA from Meta.

**Questions:**

Please see above Weaknesses.

---

### Official Review · Reviewer_xJTn · 2024-11-10

**Soundness:** 3
**Presentation:** 2
**Contribution:** 1
**Rating:** 3
**Confidence:** 4

**Summary:**

The paper introduces FeDeRA, a method for efficient fine-tuning of pre-trained language models (PLMs) in federated learning (FL) environments. FeDeRA extends the LoRA approach by initializing low-rank matrices using SVD decomposition of pre-trained weight matrices. This initialization aims to address the performance degradation of PEFT methods in FL under non-IID data conditions. The paper claims improvements in task performance, robustness to data heterogeneity, and communication efficiency, demonstrated through experiments on NLP tasks.

**Strengths:**

1. Problem Relevance: The paper addresses an important challenge: efficient fine-tuning of PLMs in federated learning under non-IID data, a setting increasingly relevant for privacy-preserving distributed learning.

2. Efficiency Gains: FeDeRA demonstrates significant reductions in communication and computational overhead while maintaining competitive task performance compared to full-parameter fine-tuning (FFT). These efficiency gains are valuable for FL settings with constrained resources.

**Weaknesses:**

1. Limited Contribution: The core contribution, using SVD for initializing the low-rank matrices, is incremental and lacks sufficient theoretical novelty. While this initialization shows empirical benefits, the idea of using SVD to improve optimization in low-rank methods is well-known in other contexts. Also, a highly relevant method (SLoRA), which overcomes the key limitations
of federated LoRA in high heterogeneous data scenarios through a data-driven initialization technique, is not compared against.
Furthermore, the remarks following equation 13, which highlight the challenges of non-IID data in federated learning, are not novel. These issues are well established in the literature, and the paper does not offer new insights or solutions beyond reiterating standard observations.

2. Weak Derivations and Inconsistent Notation: The derivations in section 4.1 lack rigor, with key variables (e.g., $L$) not clearly defined. This section is critical for understanding the challenges posed by non-IID data for PEFT in FL setting and establishing the contributions of the paper, yet it provides little beyond well-known observations. Additionally, inconsistent use of LoRA matrix multiplications (given the specified dimensions of matrices $A$ and $B$, the correct order should consistently be $BA$) reflects a lack of careful proofreading and may lead to misinterpretation of key equations.

3. Lack of Discussion on SVD Overhead: Although the paper claims SVD computation is negligible, it lacks a detailed analysis of this overhead for larger models. SVD may become computationally expensive when applied to larger pre-trained weight matrices, which could offset the efficiency benefits highlighted in the paper.

4. Baseline Comparisons and Experimental Scope: The paper’s empirical comparisons are limited to a few baseline methods (FedBF, FedAP, and FedLR). Recent competitive methods in FL and PEFT, such as SLoRA or other LoRA variants specifically designed for FL, are not included, which weakens the strength of the comparative analysis.

**Questions:**

See weaknesses.

---

### Note · Authors · 2024-11-20

**Comment:**

We sincerely appreciate the reviewers for their valuable feedback and constructive suggestions. These insights are immensely helpful, and we will carefully revise and enhance our paper based on the provided comments.

**Withdrawal Confirmation:**

I have read and agree with the venue's withdrawal policy on behalf of myself and my co-authors.